# When Care Becomes Abuse: A Forensic–Medical Perspective on Munchausen Syndrome by Proxy

**DOI:** 10.3390/pediatric17030060

**Published:** 2025-05-15

**Authors:** Emanuele Capasso, Carola Costanza, Michele Roccella, Beatrice Gallai, Michele Sorrentino, Marco Carotenuto

**Affiliations:** 1Department of Advanced Biomedical Sciences, School of Medicine and Surgery, University of Naples Federico II, 80138 Napoli, Italy; 2Department of Psychology, Educational Science and Human Movement, University of Palermo, 90128 Palermo, Italy; michele.roccella@unipa.it; 3Department of Surgical and Biomedical Sciences, University of Perugia, 06123 Perugia, Italy; beatrice.gallai@unipg.it; 4Faculty of Medicine and Surgery, UniCamillus-Saint Camillus International University of Health Sciences, 00131 Roma, Italy; michele.sorrentino@unicamillus.org; 5Clinic of Child and Adolescent Neuropsychiatry, Department of Mental Health, Physical and Preventive Medicine, University of Campania “Luigi Vanvitelli”, 80131 Naples, Italy; marco.carotenuto@unicampania.it; 6Società Italiana di Psicologia Pediatrica (SIPPed), Viale Croce Rossa n. 42, 90144 Palermo, Italy

**Keywords:** Munchausen Syndrome by Proxy, factitious disorder imposed on another, fabricated or induced illness (FII), medical child abuse, forensic psychiatry, child maltreatment, diagnosis challenges

## Abstract

Munchausen Syndrome by Proxy (MSBP) is recognized as a form of child abuse in which a caregiver induces or fabricates illnesses in their child to gain medical and social attention. MSBP represents one of the most complex and insidious forms of child abuse, characterized by an ambiguous clinical presentation that poses significant challenges for physicians, psychiatrists, and social workers. However, this condition raises critical questions regarding its diagnosis, management, and forensic implications. Traditionally, MSBP has been framed as an individual pathological manifestation of the mother, overlooking the role of the healthcare and legal systems in its identification and management. In this article, we propose a critical reflection on MSBP, emphasizing how the issue is not merely a “parental failure” but rather a systemic failure of healthcare, social, and judicial institutions in recognizing, preventing, and effectively managing such cases.

## 1. Introduction

### 1.1. Historical Roots of the Diagnosis

Munchausen Syndrome by Proxy (MSBP) is a complex form of child abuse with significant morbidity and mortality. Its manifestations include the fabrication or simulation of clinical symptoms by the caregiver and, in more severe cases, the deliberate induction of illness through acts such as poisoning, suffocation, or infections [1,2]. However, despite having been described for decades, MSBP remains a controversial phenomenon with an unclear pathogenesis. The condition can range from mild to severe forms, and its diagnosis should be recognized regardless of the caregiver’s intent [1].

In 1951, Richard Asher first described this syndrome, naming it *Munchausen Syndrome* because the affected patients reported fictitious disorders and dramatic symptoms, akin to the fantastical adventures recounted by Baron von Münchausen [3]. However, it was not until 1977 that pediatrician Roy Meadow observed this behavior not in adults, but in children. Meadow reported two emblematic cases. The first case involved a young girl presenting with a significant amount of blood in her urine, with no apparent clinical explanation. The second case described a mother who administered toxic doses of table salt to her child, leading to frequent hospitalizations and extensive diagnostic testing, ultimately resulting in the patient’s death following a temporary, apparent recovery during hospitalization [4]. He extended the concept of *Munchausen* to include caregivers who induce or falsify illness in a child, coining the term *Munchausen Syndrome by Proxy* [3,4].

MSBP is now recognized as a severe form of abuse in which a caregiver induces symptoms or injuries in an individual under their care, typically a child, an elderly person, or a disabled individual. The defining characteristic of the syndrome is that the victim is always a vulnerable person, unable to defend themselves or report the abuse.

Later, in 2002, Meadow described MSBP as a perversion of parenting, characterized by an inability to love, protect, and prioritize the child’s needs [5]. This interpretation highlights the pathological dimension of the caregiver, shifting away from a purely criminal perspective of the phenomenon. In 2013, Flaherty and Macmillan proposed replacing the term MSBP with Caregiver-Fabricated Illness (CFI) [6]. This terminology shifts the focus from the abusive parent to the victim, highlighting the need to protect the child and prevent further abuse. However, no universal consensus has been reached regarding the most appropriate terminology, and the debate over whether the label should apply to the abuse, the perpetrator, or the victim is ongoing. Since its definition, MSBP has had significant legal consequences, necessitating a thorough forensic evaluation in medico-legal investigations [7].

However, other authors have offered alternative interpretations, highlighting the potential psychiatric and social implications of the disorder.

For instance, Rosen et al. (1984) [8] investigated abusive mothers, excluding the presence of psychosis while emphasizing inadequate psychological defense mechanisms. These included a tendency to compensate for low self-esteem and an internal sense of emptiness by adopting the role of a devoted and self-sacrificing mother. This behavior, often reinforced by admiration from medical staff, can create a vicious cycle in which the mother manipulates the healthcare system to gain attention and social recognition. According to Rosen, MSBP may be rooted in Narcissistic Personality Disorder, indicating that the condition is not merely a form of child abuse but also a psychopathological disorder of the caregiver [8,9].

Despite being recognized for over three decades, MSBP remains frequently underdiagnosed or misdiagnosed by physicians. Meadow himself, in 1977, defined MSBP as “*The Hinterland of Child Abuse*”, highlighting the diagnostic challenges and ethical implications surrounding the identification of this condition [4]. In reality, MSBP constitutes a form of abuse that affects not only the child but also the healthcare system, which may unwittingly become an instrument of the abuse, subjecting the child to unnecessary and potentially harmful diagnostic and therapeutic procedures.

### 1.2. Epidemiology: An Underestimated Phenomenon

MSBP is currently considered a rare condition, with an estimated incidence of 0.5 cases per 100,000 children under the age of 16. However, this estimate is likely underreported due to diagnostic difficulties and medical professionals’ tendency not to recognize the disorder [10].

MSBP can present in various clinical forms and affect multiple medical specialties, including dermatology (self-inflicted skin lesions), endocrinology (simulated hormonal imbalances), gastroenterology (self-induced diarrhea or vomiting), neurology (factitious epileptic seizures), and infectious diseases (induced prolonged fever) [10].

A study by Sheridan et al. (2003) analyzed 451 cases from 1972 to 1999, suggesting that the incidence of MSBP is approximately 0.4 per 100,000 children aged 2 to 17 years, and 2 per 100,000 infants under 1 year old [11]. Additionally, a 1993 study reported that MSBP primarily affects children under six years old and, in severe cases, can result in death in approximately 10% of instances [12].

### 1.3. Challenges in Terminology and Diagnostic Framing

An initial attempt to define MSBP was made by Meadow, who characterized it as a phenomenon in which children undergo unnecessary and distressing hospital experiences because their parents fabricate symptoms. However, in 1987, Rosenberg expanded on this concept, establishing a fully defined clinical syndrome characterized by the following four fundamental criteria [13]:Simulation or induction of an illness in the child by the caregiver.Repeated medical evaluations and invasive procedures, often yielding no diagnostic results.Denial of responsibility by the perpetrator regarding the child’s illness.Improvement of the child’s condition upon separation from the caregiver.

One of the main difficulties in identifying Munchausen Syndrome by Proxy (MSBP) lies in the complexity of its classification and clinical framing. The DSM-5 and DSM-5-TR have replaced the traditional term with *Factitious Disorder Imposed on Another* (FDIA), classified under code 300.19 (F68.A), which emphasizes the intentional deception by a caregiver without external incentives [14]. This condition, now grouped under Somatic Symptoms and Related Disorders in both the DSM and the ICD-11 (6D.51), may remain undetected for long periods due to its subtle presentation and the lack of specific biological markers [15].

Clinicians often rely on caregiver reports without questioning their accuracy, especially when the caregiver presents as attentive or concerned. This over-reliance can lead to repeated and unnecessary diagnostic procedures, potentially causing harm to the child and delaying appropriate intervention [14]. Although MSBP remains a familiar term in clinical settings, more recent alternatives—such as *Medical Child Abuse* and *Fabricated or Induced Illness in a Child*—aim to shift the focus from caregiver motivation to the harm inflicted on the child, promoting clearer identification and interprofessional collaboration [15].

The term Medical Child Abuse (MCA) emphasizes the harm caused to the child through unnecessary medical interventions, whether by fabrication or induction, rather than focusing on the caregiver’s presumed psychological motivations. This terminology facilitates interdisciplinary collaboration by allowing pediatricians, child psychiatrists, and neuropsychiatrists to work together. As highlighted in the recent literature [16,17], MCA offers a practical framework for identification, reporting, and intervention in such cases.

### 1.4. Diagnostic Implications Criteria

Rosenberg’s criteria remain a useful framework for recognizing FDIA. These include repeated healthcare visits without a clear etiology, test results suggesting manipulation, clinical presentations not attributable to medical error, and a lack of plausible medical explanations [13,18]. More recently, direct indicators such as inconsistencies between caregiver narratives and clinical findings, falsified documents, or surveillance evidence have gained importance as objective diagnostic tools.

Because diagnostic mistakes can lead either to unjust accusations or missed cases of abuse, a multidisciplinary approach is considered essential [19,20]. Evidence from a 1990s UK-based study supports the value of an active pediatric surveillance system that incorporates dedicated assessment units, child protection meetings, detailed pediatrician reports, and judicial involvement to reduce diagnostic errors and ensure child safety [19]

Some clinical reports have described variability in the pattern of caregiver behaviors, ranging from the persistent and repeated induction or falsification of symptoms (long-term pattern), to more discrete, situationally triggered episodes [6]. These distinctions aim to describe behavioral patterns without implying specific psychological motives, which are often difficult to verify and are outside the scope of pediatric assessment. In severe cases, the child may suffer from real physical injuries, with symptoms ranging from hyperglycemia to diarrhea. In some instances, mothers may manipulate biological samples, falsifying diagnostic tests [20]. Moreover, MSBP may begin with an actual illness in the child, which the caregiver artificially worsens to prolong medical attention [21,22].

A peculiar aspect of MSBP is how caregivers respond to medical suspicion: some threaten legal action against healthcare professionals, and others may escalate the abuse to convince physicians of the severity of the child’s condition.

In this context, awareness among healthcare professionals is crucial, particularly in an era where defensive medicine can lead to misdiagnoses [23]. The deceptive caregiving behavior characteristic of FDIA can severely compromise the child’s ability to establish secure attachments. The contradiction between receiving care and simultaneously being harmed by the caregiver fosters confusion, mistrust, and emotional dysregulation, which may persist long into adulthood. There are severe consequences for the child, who may develop post-traumatic stress disorder (PTSD), anxiety, and issues related to identity.

### 1.5. Caregiver Behavioral Profile: From Classic Criteria to a Modern Identikit

Numerous studies have sought to identify the psychological dynamics and personality disorders frequently linked to the caregiver’s pathological behavior, suggesting that child maltreatment may serve as a means to stabilize a dysfunctional family. In terms of the role of gender, 95% of MSBP cases are perpetrated by women, primarily mothers [24]. The link between female roles and psychological distress may elucidate this prevalence. Women, more than men, form an identity centered on caregiving, and in some instances, their pathological need for attention may lead them to exploit their children [25,26]. Although the majority of documented perpetrators are mothers, several reports in the international literature have described cases involving fathers or other male caregivers. Including these examples broadens our understanding of the phenomenon and reinforces the need for thorough and unbiased investigations [11].

A 2015 study [27] identified common traits among abusive mothers, including extreme dedication to their child, extensive medical knowledge, interference with medical care, a lack of natural composure in the face of serious medical difficulties, and a preference for the hospital environment. In many cases, the father is a passive figure, distant and uninvolved [28]. In a study analyzing 796 cases, 30% of perpetrators had a factitious disorder, while 18.6% had a personality disorder [24]. These findings suggest that MSBP is a complex condition deeply rooted in psychopathological and social dynamics [29,30]. The psychiatric condition of the perpetrator of MSBP [31] may exhibit underlying personality disorders, mood disorders, or other psychiatric conditions that do not necessarily fall under the classification of Factitious Disorder Imposed on Another [32]. This distinction carries prognostic and therapeutic implications that cannot be sufficiently addressed solely through a diagnostic label. A summary of key clinical, behavioral, and psychopathological characteristics of the FDIA caregiver, integrating classical and contemporary sources, is provided in Table 1.

## 2. Legal and Forensic Considerations in FDIA Cases

Due to its multifaceted nature, Munchausen Syndrome by Proxy (MSBP) requires careful examination not only from a clinical standpoint but also through legal and forensic lenses. One of the primary legal challenges lies in the precise identification of the offense and the appropriate attribution of responsibility. Sigal et al. (1990) observed that invoking MSBP as a formal psychiatric diagnosis in criminal defense strategies may facilitate arguments based on diminished responsibility [15]. However, Rand and Feldman have cautioned that the application of the MSBP label to a caregiver or child increases the likelihood of diagnostic errors, particularly when the condition is viewed as a clinical syndrome rather than a specific pattern of abuse [31].

A well-supported and conclusive diagnosis is not only a prerequisite for legal justice but also a necessary safeguard for potentially at-risk individuals, especially siblings of the victim, who may be silently exposed to similar abuse. The lack of standardized and universally recognized diagnostic criteria creates the dual risk of failing to protect truly abused children while simultaneously increasing the chances of false accusations against innocent caregivers [33].

An emerging concern in the literature is the role of digital platforms in shaping new forms of caregiver deception. Recent studies have highlighted the phenomenon known as “Munchausen by Internet”, where parents exploit social media to construct narratives of non-existent childhood illnesses in order to gain emotional sympathy or financial support from online communities [34]. This digital evolution of FDIA complicates detection, as the abuse often manifests through shared content rather than direct physical harm, blurring the boundaries of clinical and legal evidence.

Compounding these difficulties is the current absence of standardized forensic protocols for establishing FDIA as a cause of death in post-mortem evaluations. This diagnostic void is largely due to the heterogeneity of clinical presentations, which continues to obstruct definitive conclusions in suspected fatal cases.

### 2.1. Strategies

#### 2.1.1. Post-Mortem Diagnosis

When a child dies in circumstances that raise clinical or forensic suspicion, particularly where Munchausen Syndrome by Proxy (MSBP) may have played a role, the post-mortem examination becomes a fundamental tool in determining both the cause and the manner of death [35]. The autopsy must be accompanied by a thorough and targeted toxicological analysis. This includes the examination of fluids such as vitreous humor, blood, urine, and gastric contents, with the goal of identifying toxic substances or medications that were not prescribed or indicated. In suspected cases of salt poisoning, for example, it is essential to perform specific laboratory comparisons between serum and urinary sodium levels to confirm or rule out intentional induction [36]. Additionally, microbiological cultures are crucial in distinguishing genuine infections from post-mortem contamination, a frequent confounder in these settings [37,38]. Careful attention should also be paid to the analysis and preservation of any implanted medical devices—such as catheters, shunts, or endotracheal tubes—which may show signs of tampering or deliberate obstruction.

#### 2.1.2. In Vivo Diagnosis

The identification of MSBP in living patients remains extremely complex, requiring a combination of clinical intuition, interdisciplinary collaboration, and objective evidence. A detailed review of the child’s medical history is often the first step, with attention paid to recurrent hospital admissions, patterns of unexplained symptoms, and invasive treatments not justified by clinical findings. Covert video surveillance (CVS), where legally and ethically permissible, can provide direct evidence of harmful behavior by the caregiver. In some cases, a temporary separation of the child from the suspected perpetrator has led to a rapid and spontaneous improvement in the child’s condition, offering powerful clinical confirmation of caregiver-induced illness.

Beyond observational strategies, targeted laboratory investigations play a pivotal role. Toxicological screening may reveal substances responsible for poisoning [39], while blood group and DNA testing can help identify contamination or substitution of samples [40,41]. More specialized assays include the detection of ipecac alkaloids in cases where vomiting has been induced [40,42], the use of radioactive chromium (51 Cr) to expose fictitious blood transfusions [43], and the analysis of sweat potassium levels to uncover fraudulent cystic fibrosis diagnoses [44]. In cases of fabricated metabolic disorders, identifying ascorbic acid in urine may point to false-positive results for diabetes mellitus [45], while abnormal insulin and C-peptide levels can suggest the artificial induction of hypoglycemia in otherwise healthy children [46].

## 3. Conclusions and Future Implications

Munchausen Syndrome by Proxy (MSBP) can no longer be viewed merely as a psychiatric disorder confined to the individual; instead, it should be addressed as a systemic concern that encompasses medicine, the justice system, and social services [47]. To enhance the management of this condition, it is essential to conduct the following:

Establish objective diagnostic protocols in both clinical and forensic settings; improve the training of healthcare professionals to enable them to recognize suspected cases without elevating the risk of false positives; and strengthen collaboration among physicians, psychiatrists, social workers, and legal experts to ensure adequate protection for victims.

In addition, it is essential to create specialized units for case surveillance, featuring multidisciplinary teams adept at conducting thorough evaluations; and implement preventive strategies, intervening in at-risk families before abuse occurs. Ultimately, enhanced interdisciplinary dialog could help balance child protection with parental rights, preventing judicial errors that might either ruin innocent families or leave children vulnerable to ongoing abuse. The effective management of FDIA cases should not only aim to protect the child but also include structured psychological intervention for both victims and perpetrators. While children may benefit from trauma-informed therapy to restore emotional safety and autonomy, caregivers require tailored psychiatric assessment and, when appropriate, forensic treatment pathways addressing underlying psychopathology [48]. Finally, the need for psychological intervention must extend beyond the child victim to include the caregiver, especially in cases where underlying psychiatric disorders are suspected. Tailored therapeutic approaches, in combination with forensic assessments, are essential both for preventing recurrence and for informing judicial decisions. As Sanders and Bursch highlight, a multidisciplinary psychiatric evaluation of the caregiver is crucial to address the psychopathology behind the abuse and to define appropriate legal and clinical responses [46].

## Figures and Tables

**Table 1 pediatrrep-17-00060-t001:** Clinical, behavioral, and psychopathological profile of the caregiver. This table synthesizes the key traits and indicators associated with Factitious Disorder Imposed on Another (FDIA), drawing from classical diagnostic frameworks (Rosenberg, 1987) [13] and more recent behavioral and forensic observations.

Category	Key Features	Clinical/Forensic Relevance
Symptom Induction	Fabrication or induction of illness, often via medication or tampering.	Requires toxicology, observation, or surveillance.
Medical Behavior	Frequent, unexplained hospital visits; insistence on interventions.	Suggests pattern of medical overuse and manipulation.
Diagnostic Clues	Inconsistencies in history; falsified samples or records.	Objective basis for suspecting FDIA.
Response to Suspicion	Threats to professionals; symptom escalation.	Signals defensive manipulation.
Psychodynamic Traits	Low self-esteem; emotional void; attention-seeking caregiving identity.	May reflect narcissistic traits (Rosen et al., 1983) [8].
Observed Patterns	Over-identification with medical staff; extensive medical knowledge; excessive devotion to the child; interference with medical care; unnatural calm in critical situations; preference for clinical environments.	Ozdemir et al. (2015) [27].

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
