# Peer review of "When Care Becomes Abuse: A Forensic–Medical Perspective on Munchausen Syndrome by Proxy"

_pediatrrep, 2025, doi:10.3390/pediatric17030060_

Round 1
Reviewer 1 Report
Comments and Suggestions for Authors
Dear authors, thank you for your manuscript, which provides an overview of the important topic of what used to be called Munchhausen Syndrome by proxy. I agree with many of your statements, but as a pediatrician, I see the core dilemma underlying the problem reflected in your article: is it a form of child abuse or a parental disorder? To me, this is one of the main reasons why this entity presents diagnostic challenges - clinicians who treat children are not the ones who are trained to assess the mental health of parents. In line 161 you define the chronic form by a parental motivation ("seeking attention"), while the episodic form is defined by parental actions ("induces symptoms"). This mixture of actions and speculation about their motivation is part of the diagnostic problem and illustrates why the term Munchausen by proxy is rather counterproductive.
However, I do not see how your article offers a solution to this dilemma.
And there indeed are proposals to resolve this dilemma by redefining and broadening the definition with a clear focus on the harm being done to the child: the definition of medical child abuse 1,2. You mention this among other terms without going into detail. Paragraph 1.3 reads like a list without providing context and without organizing the facts for the reader.
I disagree with the point you make in line 127: medical child abuse is not a diagnosis of exclusion alone. Deliberate concealment and misrepresentation by parents can be objectified.
The statement in line 154 lacks a reference.
1 Hornor G. Medical Child Abuse: Essentials for Pediatric Health Care Providers. J Pediatr Health Care. 2021 Nov-Dec;35(6):644-650. doi: 10.1016/j.pedhc.2021.01.006. Epub 2021 Feb 12. PMID: 33589306.
2 Semrau GM, Härlin R, Di Maria C, Schwartländer B, Winter SM. Medical child abuse - a guideline for the diagnosis of this special form of emotional and physical violence. Prax Kinderpsychol Kinderpsychiatr. 2024 J
Author Response
We sincerely thank the Reviewer for the constructive and insightful comments, which helped us significantly improve the clarity, coherence, and scientific accuracy of the manuscript. Thanks to your thoughtful suggestions, we believe the revised paper is now more robust and balanced. All modifications in the text are highlighted in yellow for clarity.
Comment (line 161): You define the chronic form by a parental motivation (“seeking attention”), while the episodic form is defined by parental actions (“induces symptoms”).
Response: We appreciate this observation and fully agree. The previous dichotomy between chronic and episodic forms reflected an outdated academic framing, which risked introducing confusion by mixing observable behavior with inferred psychological motives. In the revised manuscript, we replaced this with a formulation grounded solely in observable caregiver actions, without speculative intent. The updated version, now highlighted in yellow in the text, reads: “Some clinical reports have described variability in the pattern of caregiver behaviors, ranging from persistent and repeated induction or falsification of symptoms (long-term pattern), to more discrete, situationally-triggered episodes. These distinctions aim to describe behavioral patterns without implying specific psychological motives, which are often difficult to verify and outside the scope of pediatric assessment.”
Comment: I do not see how your article offers a solution to this dilemma.
Response: We acknowledge this point and have incorporated a more explicit position. In the revised manuscript, we propose addressing the diagnostic ambiguity by adopting more punctual descriptive terminology. Specifically, we advocate for the use of the term Medical Child Abuse, which helps separate the caregiver's psychiatric profile from the harm inflicted on the child. This terminological shift now explained in a dedicated section (highlighted in yellow), emphasizes clinical and forensic evidence over inferred psychological intent.
Comment: You mention the term Medical Child Abuse among others, without going into detail.
Response:Thank you. We have expanded paragraph 1.3 to include a detailed explanation of Medical Child Abuse, referencing both Hornor (2021) and Semrau et al. (2024), as you suggested. This section now better contextualizes and supports the terminological shift discussed above.
Comment (paragraph 1.3): It reads like a list without context.
Response: We agree. The paragraph has been rewritten to form a cohesive narrative that integrates diagnostic and terminological challenges with greater clarity and flow. The revised text is highlighted in yellow in the manuscript.
Comment (line 127): Medical Child Abuse is not a diagnosis of exclusion alone.
Just to let you know, we agree. The text has been revised to reflect that Medical Child Abuse can be supported by objective findings . It is no longer described as solely a diagnosis of exclusion.
Comment (line 154): Missing reference.
Response: Thank you for noticing. The missing reference has been added and is now properly cited in the text.
Reviewer 2 Report
Comments and Suggestions for Authors
I thank the authors for introducing this review of the literature and their absolutely concurring reflection of how the syndrome is also a failure of the health care system. I have only once encountered a case of MS, and medical observations and police investigations were instrumental in reaching the truth of the facts and placing the child in protection.
Here are some points that could be considered in your review:
- Point out some history of why it is really called Munchausen Syndrome.
- it is true that overwhelming majority of the perpetrators are mothers, but I would reinforce the construct that there are also father perpetrators, as a few cases have been recounted in the international literature.
- Refer to the possible repercussions on attachment.
- what kind of psychological intervention for perpetrators and victims of the syndrome?
Author Response
We are grateful for your thoughtful comments and your personal reflection on the complexity of MSBP/FDIA cases. Your suggestions have been extremely helpful, and we have addressed each of your points in the revised manuscript, with all modifications clearly highlighted in yellow:
-
A brief historical note has been added to Section 1.1, outlining the origins of the term Munchausen Syndrome as introduced by Asher in 1951, and its subsequent application to proxy cases by Meadow in 1977.
-
While it is true that the vast majority of documented perpetrators are mothers, we have now clarified that fathers have also been identified as perpetrators in several international reports. This revision was made to ensure a more inclusive and evidence-based framing.
-
We have incorporated a brief discussion of the potential repercussions on the child's attachment system, underlining how deceptive and manipulative caregiving may impair the development of secure relational bonds.
-
Lastly, the concluding section has been expanded to emphasize the need for psychological interventions not only for the child victim but also for the caregiver. This includes a reference to both therapeutic and forensic approaches where applicable.
In particular, we now state: “Finally, the need for psychological intervention must extend beyond the child victim to include the caregiver, especially in cases where underlying psychiatric disorders are suspected. Tailored therapeutic approaches, in combination with forensic assessments, are essential both for preventing recurrence and for informing judicial decisions. As Sanders and Bursch highlight, multidisciplinary psychiatric evaluation of the caregiver is crucial to address the psychopathology behind the abuse and to define appropriate legal and clinical responses [47].”
Thank you once again for your constructive feedback, which has helped us strengthen the manuscript in both depth and clinical relevance.
Reviewer 3 Report
Comments and Suggestions for Authors
The manuscript addresses a highly relevant and underexplored area of child abuse—Munchausen Syndrome by Proxy (MSBP)—through a multidisciplinary and forensic-medical lens. The introduction is comprehensive and well-cited, offering valuable historical context and contemporary relevance. The manuscript succeeds in emphasizing the systemic failures surrounding MSBP, moving beyond a narrow clinical diagnosis to include judicial and institutional dynamics. The discussion of diagnostic challenges and forensic implications is timely, especially with the emergence of digital variants such as “Munchausen by Internet.” Your integration of psychiatric, legal, and clinical dimensions presents a holistic view that will be of interest to both medical and legal professionals.
Suggestions for improvement:
- While the content is rich, reorganizing certain sections (for example, separating out forensic from psychiatric considerations more clearly) may improve flow and reader navigation.
- While the conclusions are insightful, consider briefly suggesting specific practical frameworks or protocols that could be implemented in clinical and forensic settings.
- A table summarizing the diagnostic criteria, clinical subtypes, or differences between in vivo and post-mortem indicators might help reinforce key points for readers.
- Some sections would benefit from careful language editing to improve clarity and readability. For example, certain sentences are overly long or include awkward phrasing. A professional proofreading round is recommended
Author Response
We sincerely thank the reviewer for the generous and insightful evaluation of our manuscript. We are particularly pleased that the multidisciplinary and forensic orientation of the work was recognized as both relevant and timely. Your comments have contributed significantly to refining the structure and content of the manuscript.
In response to your suggestions:
-
We have reorganized specific sections to ensure a clearer distinction between psychiatric considerations and forensic aspects, thereby improving both narrative coherence and reader navigation.
-
In the revised conclusion, we have added a brief proposal outlining practical implementation strategies in clinical and forensic contexts. These suggestions are informed by recent literature and professional practice frameworks.
-
A new summary table has been included to highlight key diagnostic criteria and caregiver behavioral indicators, serving as a practical synthesis for clinicians and forensic professionals. (Table 1)
-
We appreciate your recommendation regarding language quality. The manuscript has been carefully revised for clarity and fluency, and we are arranging a final round of professional proofreading prior to submission.
All modifications in response to your comments are highlighted in yellow in the revised manuscript.
Thank you once again for your valuable feedback, which has helped improve the manuscript both in substance and presentation.
Round 2
Reviewer 1 Report
Comments and Suggestions for Authors
Thank you for the revision of the manuscript. In this improved version, I do recommend the publication in Pediatric Reports.